# Use of Neural Networks for Tsunami Maximum Height and Arrival Time Predictions

**Juan F. Rodríguez** *[ID], **Jorge Macías** [ID], **Manuel J. Castro** [ID], **Marc de la Asunción** and **Carlos Sánchez-Linares** [ID]

Departamento de Análisis Matemático, Estadística e Investigación Operativa y Matemática Aplicada, Campus de Teatinos, s/n, University of Málaga, 29071 Málaga, Spain; jmacias@uma.es (J.M.); mjcastro@uma.es (M.J.C.); marcah@uma.es (M.d.l.A.); csl@uma.es (C.S.-L.)
* Correspondence: juanrg@uma.es

**Abstract:** Operational TEWS play a key role in reducing tsunami impact on populated coastal areas around the world in the event of an earthquake-generated tsunami. Traditionally, these systems in the NEAM region have relied on the implementation of *decision matrices*. The very short arrival times of the tsunami waves from generation to impact in this region have made it not possible to use real-time on-the-fly simulations to produce more accurate alert levels. In these cases, when time restriction is so demanding, an alternative to the use of decision matrices is the use of datasets of precomputed tsunami scenarios. In this paper we propose the use of neural networks to predict the tsunami maximum height and arrival time in the context of TEWS. Different neural networks were trained to solve these problems. Additionally, ensemble techniques were used to obtain better results.

**Keywords:** tsunami modeling; deep learning; neural network; maximum height; arrival time

## 1. Introduction

The impact of a tsunami can be devastating, both on human lives and economically. Some recent works [1] show us how the impact of tsunamis has increased in the last few years, with a total of 251,770 casualties and USD 280 billion in damages between 1998 and 2017. For this reason, among others, there exist tsunami early warning systems (TEWS), and it is of key importance to continuously improve them.

The two main variables these systems must try to accurately anticipate are the maximum height of the incident waves and the arrival time of those. In this work, we will discuss how by taking advantage of deep learning techniques we can provide extremely fast and accurate predictions of these two variables.

Nowadays, there exist very efficient computer codes, implemented in massive parallel architectures as GPUs (graphics processing units), that return very accurate results in quite short computing times. Nevertheless, the computational resources required to include the variability associated with the uncertainty in the seismic source are still quite large and national TEWS do not have these resources available. In addition, even for a single "most probable" or "worst-case" scenario, depending on the level of conservatism, we may want to drastically reduce the time-to-solution. Here is where machine learning, or more concretely, deep learning, come into action. However, neural networks (NN) require large amounts of data to be trained, and regarding tsunamis, which are rare events, few data are available from natural events. Therefore, we lack the first main ingredient for building the model. Nevertheless, we can use our traditional computer tsunami models to generate the huge amount of data required to train NN. In addition, we can combine several NN models to produce more accurate results. Once the NN model is trained, it can be used to predict the maximum height and the arrival time for a particular event in just a few seconds.

Machine learning (ML) techniques are quickly spreading and being used in all fields of research, and tsunami science is not an exception. Many authors are starting to use ML techniques for tsunamis ([2–6]).

In this paper, multi-layer perceptron (MLP) neural networks [7] will be used to forecast tsunami maximum height and arrival time at certain points on the Chipiona-Cádiz coast (southwestern Spain). First, we considered single models and they produced good results. Next, in order to improve the results of those single models, ensemble techniques were also considered and implemented to reduce the variance of the results obtaining, in general, better predictions.

The present work is organized as follows: Section 2 describes the context in which the present work arises and the simulations that will serve to train the NN models; Section 3 briefly describes the NN models to be used; the single NN models proposed are described in Section 4 and the numerical results obtained using single and ensemble models are presented in Section 5; Section 6 briefly describes how the basic tool proposed in this work could be used as building blocks of a real operational TEWS; and finally Section 7 is devoted to highlighting some concluding remarks.

## 2. Motivation and Training Data

The aim of this work is to generate alert levels for a tsunami event in a very short time (seconds). Nowadays, in TEWS in Europe, this is still performed using decision matrices, a rudimentary approach, not considering local effects, nonlinearities of the phenomena, or even large masses of land preventing the tsunami wave reaching certain regions. In order to include a numerical approach in these systems, databases of precomputed scenarios are generated. When an event occurs, the system searches for a best-fit scenario in the database to be used as a model for the actual event. The main disadvantage of such a system relies on unexpected events not included in the database. Finally, few systems include FTRT (faster than real time) tsunami simulations on-the-fly of one or several deterministic scenarios produced in few minutes. This is achieved at IGN in Spain or CAT-INGV in Italy using the Tsunami-HySEA code.

Here, we propose to use ML techniques and NN models to generate the alert levels in few seconds. This paper will work on two regression problems. They consist of one predicting the maximum water height, and the other one predicting the arrival time of a tsunami at certain points in the coast. Here, we consider the Chipiona-Cádiz coast (segment CA01 in Figure 1).

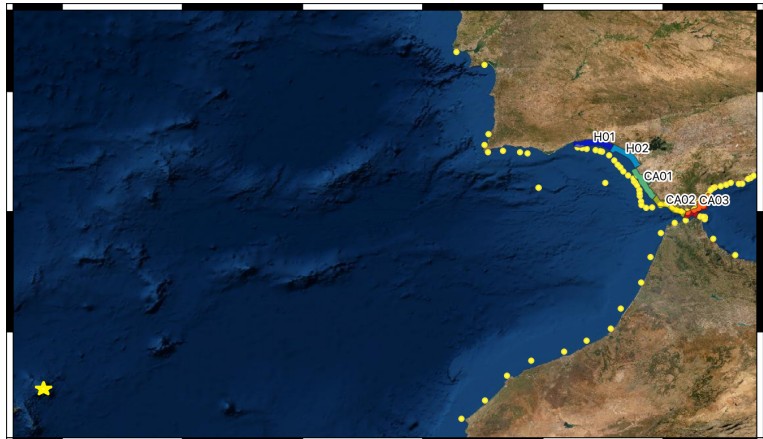

**Figure 1.** Points and coastal zones of interest. The ★ marks the approximate location of the center of the Horseshoe fault. The coast is divided into 5 sections, two for Huelva coast and three for Cádiz. We focus on CA01. Yellow dots are the virtual gauges considered by the Spanish TEWS.

The Tsunami-HySEA code, developed by the EDANYA group at the University of Málaga, Spain, was used for the tsunami simulations [8,9]. The Okada model [10] is used to compute the initial seafloor deformation caused by the earthquake. Tsunami-HySEA is currently being used in the TEWS of several countries (Spain, Italy, Chile), and it has been widely tested and benchmarked [9,11,12].

For this study, we selected the Horseshoe fault as the seismic source, that is, the fault where the earthquake generating the tsunami is produced. This means that we are designing an NN model for tsunamis generated by this particular fault. As the aim of the present study is to define a methodological approach that can be used in any marine region with the potential of generating tsunamis, the method proposed here would have to be replicated for each active fault in the area of study in order to cover all the potential seismic tsunamigenic sources. Therefore, the choice of the particular fault we use in the present study is, in some sense, not really important. In any case, we chose the Horseshoe fault because it is supposed to be the source for the major event in the region, the Lisbon 1755 earthquake and tsunami [13].

In Table 1, we have listed the Okada model parameters and specified the range of values considered for each of the parameters. To generate the actual values within this range used to produce the numerical simulations, a Sobol sequence [14] with 16,000 samples was used. Then, the Tsunami-HySEA model is used to simulate the 16,000 scenarios.

**Table 1.** Range of values for the Okada model parameters used for generating the numerical simulations used to train the NN models.

| Okada Parameter | Range of Values |
| --- | --- |
| Longitude | $[-10.3854, -9.3854]$ |
| Latitude | $[35.5192, 36.5192]$ |
| Depth (km) | $[8.5, 14.5]$ |
| Length (km) | $[89.0, 129.0]$ |
| Width (km) | $[36.0, 56.0]$ |
| Strike (degrees) | $[43.0, 53.0]$ |
| Dip (degrees) | $[20.0, 40.0]$ |
| Rake (degrees) | $[80.0, 100.0]$ |
| Slip (m) | $[3.05, 4.95]$ |

In order to have a finer resolution at the coast, the simulations were performed using three levels of nested grids. Grid level 0 has a spatial resolution of 320 m and 2,981,518 volumes, grid level 1 is 160 m and has 701,668 volumes, and grid level 2 is 40 m and is composed of 1,277,952 volumes. The size of the problem to be solved is 4,961,138 volumes in three spatial resolutions, and every single simulation takes approximately 280 s in a single GPU NVIDIA V100. This means that 1245 GPU computing hours are required. If we have access to a modest 64 GPU cluster, this can be computed in less than 20 h. If we have dedicated access to the whole MARCONI100 at CINECA, the Italian Supercomputer Center, (with 800 nodes and 3200 V100 GPUs, this would take less than 25 min.

For the present work, we selected six points, located on the segment of the coast named CA01 in Figure 1. The maximum wave height and the arrival time of the first wave are going to be predicted by the NN model in these six points. They are located close to Sanlúcar de Barrameda, Chipiona, Arriates, the Arroyo Hondo river mouth, Rota, and Cádiz (see Figure 2). The geographical coordinates for these points and the model-interpolated depths are collected in Table 2.

We made the assumption that the tsunami arrives to a certain cell when the water level differs by 0.5 cm at that cell with respect to the initial water level, that is, if $|\eta - \eta_0| \geqslant 0.5$ cm, where $\eta$ and $\eta_0$ are the current and the initial water level, respectively.

**Table 2.** Geographical coordinates of the locations used as forecasting points. (*) Depth values (in meters) refer to the numerical interpolations.

| Location | Longitude | Latitude | Depth ($*$) |
|---|---|---|---|
| S. Barrameda | −6.3601° | 36.7858° | 0.1934 |
| Chipiona | −6.4415° | 36.7321° | 1.3131 |
| Arriates | −6.4326° | 36.7032° | 1.4527 |
| A. Hondo | −6.4119° | 36.6735° | 0.5289 |
| Rota | −6.3605° | 36.6161° | 1.2638 |
| Cádiz | −6.3083° | 36.5339° | 0.8280 |

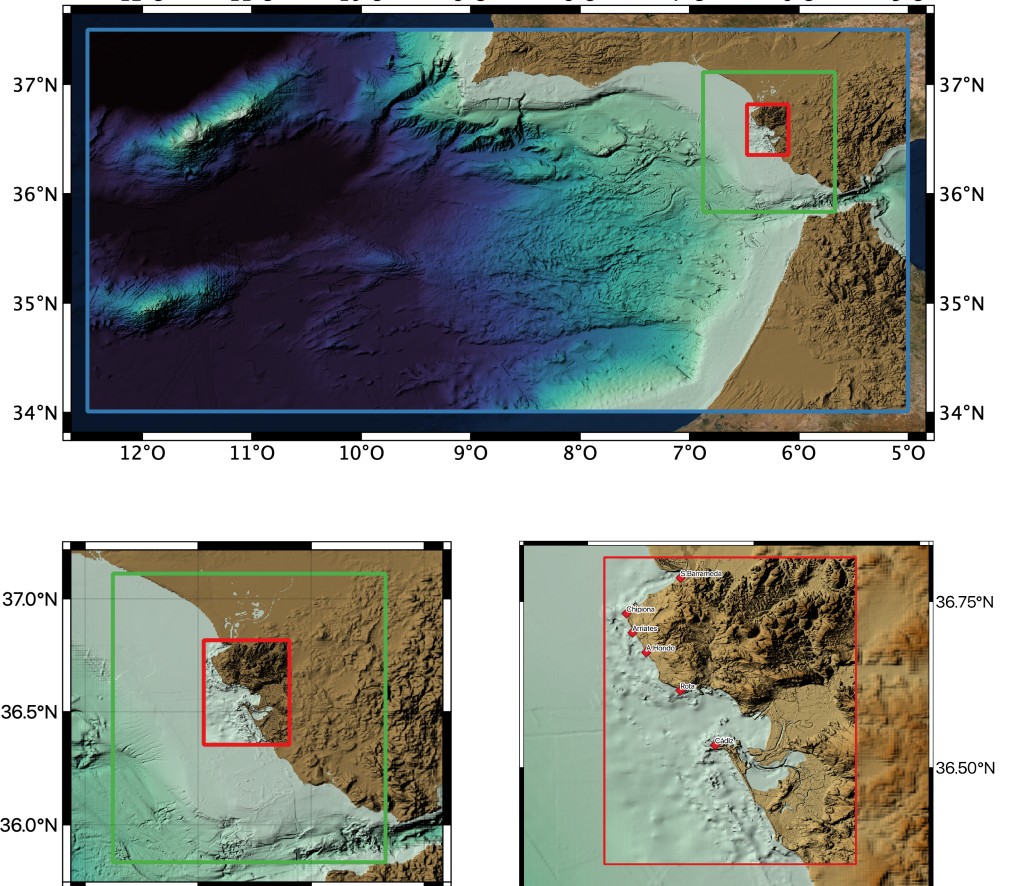

**Figure 2.** Bathymetry and extent of grid level 0 (**top panel**), grid level 1 (**left bottom panel**), and grid level 2 (**right bottom panel**).

In order to visualize the numerical results that are obtained for any of the large number of numerical simulations that we performed, we considered a particular seismic event, defined by the parameters in Table 3. This scenario plays no particular role in the construction of the NN models presented in this work. We consider it to show the outputs extracted from each simulation used to train the NN models. In addition, in Figure 3, the time series for the water height at the six forecast locations for this event are presented. Values for the time series were extracted every 5 s during 4 h.

**Table 3.** Okada parameters for the reference scenario considered. Depth, length, and width are in km, and slip in m.

| Okada Parameters | | | | | | | | |
|---|---|---|---|---|---|---|---|---|
| **Longitude** | **Latitude** | **Depth** | **Length** | **Width** | **Strike** | **Dip** | **Rake** | **Slip** |
| −9.8854° | 36.0192° | 11.5 | 109.0 | 46.0 | 48.0° | 30.0° | 90.0° | 4.0 |

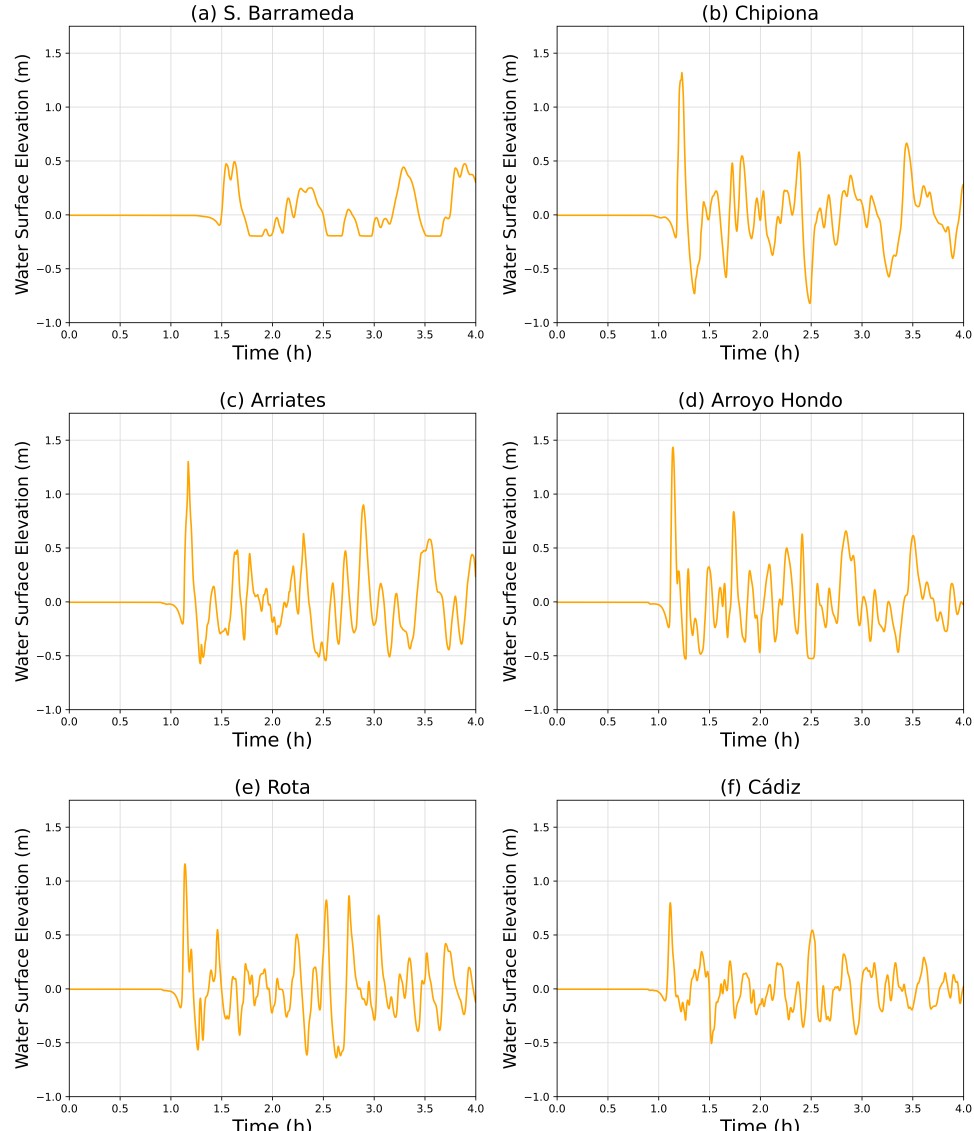

**Figure 3.** Time series for the sea surface elevation at the six forecast points for the reference event.

Different regression techniques were implemented and tested, such as decision trees or random forest. Nevertheless, the predictions were not good enough to retain these models. Therefore, deep learning was used to predict the maximum height and arrival times in the minimum time possible, obtaining, at the same time, a very good accuracy.

## 3. Neural Network and Ensemble Learning

Before presenting the forecast obtained with the NN, a brief explanation of how a neural network works, its structure, what the ensemble used for training is, why it works well in practice, and what type of ensemble has been used will be given. Nevertheless, a discussion about the different types of activation functions, back-propagation, error functions,

optimizers, regularizations, callbacks, etc., will be not given. More information about these concepts can be found in [15–18].

MLP neural networks were used to address the two problems considered in the present study. The type of neural networks that we used have a similar structure to the one shown in Figure 4, but with more hidden layers and more neurons per hidden layer. We can think of the $x_i$ as the Okada parameters (normalized to [0,1]), and $\hat{y}_i$ as the results of each sample once forward-propagation is applied. The $\hat{y}_i$ values will be changing thanks to the use of back-propagation [16]. When the last epoch is over, these $\hat{y}_i$ will be the result returned by the trained neural network. The term epoch refers to the number of passes the algorithm has completed through the entire training dataset.

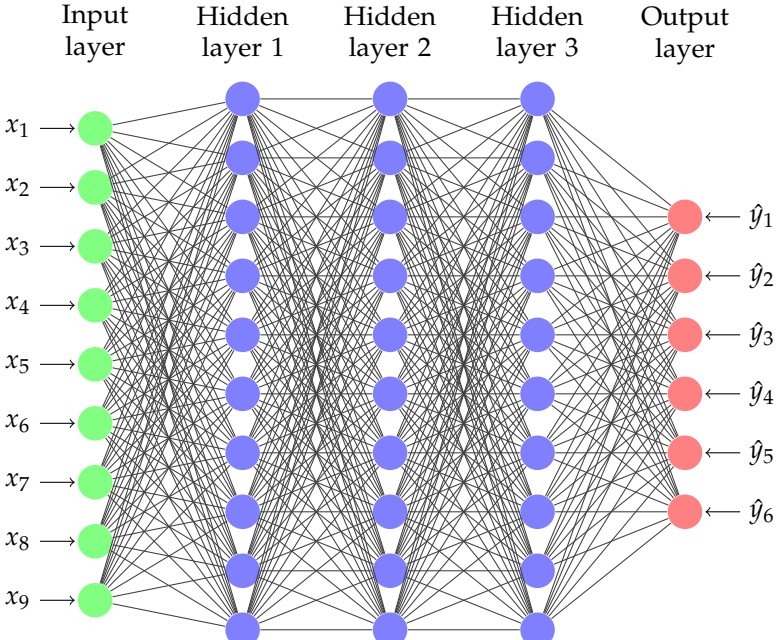

**Figure 4.** Schematic structure of an MLP neural network similar to the one used in this study.

As we can see, neural networks work similar to a human neuron; they receive the data, they process it, and, finally, a result is returned, but how does the neural network process the data? It takes two steps per layer. First, it combines, linearly, the output of the previous layer (input of the actual layer) and the synaptic weights, $w_{j,i}^{(l)}$. These weights are the values that the network will learn trying to find the best prediction. The resulting value will be denoted by $z$, and it is called *net income*. In the second step, an activation function [17], $\sigma$, is applied to the net income in order to determine the output of the layer. The activation functions are the ones which remove the linearity. Each layer has their own activation function. Visually, it can be seen in Figure 5.

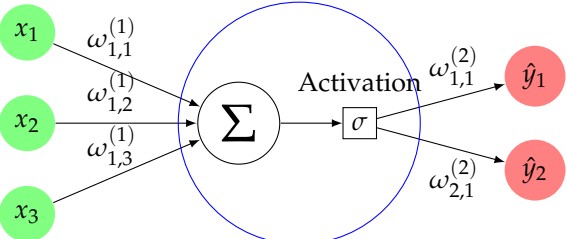

**Figure 5.** Process in each layer of the neural network. Step 1: linear combination. Step 2: application of the activation function.

Therefore, defining as $w^l$ the matrix of weights that connect to the layer $l$, the net income, $z^l$, and the output, $\hat{y}^l$, the output can be written matricially as

$$\hat{y}^l = \sigma(z^l) = \sigma\left(w^l z^{l-1}\right).$$

If we have several neural networks trained to solve the same problem, we could think to combine them to obtain better predictions. This is called *ensemble learning*. Ensemble learning follows the principle of "the wisdom of the crowd", which shows that a large group of people with average knowledge on a topic can provide reliable answers to questions such as predicting quantities, spatial reasoning, and general knowledge. The aggregate results cancel out the noise and can often be better than those of highly trained systems. Ensemble learning combines several individual models to obtain better generalization performance. The reason why ensemble learning is efficient is because each model obtains different results. Each model might perform well on some data and less accurately on others. Variance is reduced to make better predictions.

There are many types of ensemble methods [19,20]. In our case, we used a weighted ensemble, which allows us to set for each model a weight (the contribution of the model in the ensemble). We need to define a list of new weights, $\alpha$ (they are different from the weights of the neural network we mentioned above). This list is filled with values we can give manually (grid search) or randomly (random search).

$$\alpha = [\alpha_a, \alpha_b, \dots, \alpha_N]$$

The elements of the list $\alpha$ are combined with all, considering all the possible $n$-tuples, where $n$ is the number of models in our ensemble. Then, these $n$-tuples are normalized to $[0, 1]$. We have lots of possibilities, specifically, $N^n$, where $N$ is the dimension of vector $\alpha$. We can choose an arbitrary $N$, taking into account that a higher value will have a higher computational cost. As observation, it could happen that all weights are 0, then we have to discard this option. Thus, if we have $n$ models, $M_i$, $i = 1, 2, \dots, n$, and we denote by $E$ the ensemble learning, it writes as follows:

$$E = \sum_{i=1}^{n} \alpha_i R_i$$

where $R_i$ is the obtained result from model $M_i$, and $\alpha_i$ is the weight, once normalized, associated with the results of model $M_i$.

## 4. A First Approach: Single Neural Network Models

In this section, some neural network models that were trained, and that are similar to the selected final model used in this work, are described. Other techniques were tried too, producing worse results. We show some samples from networks with good results, where we can also observe the influence of certain parameters. The section is divided in two parts, one for the tsunami maximum height problem and another for the arrival times problem. We use the infinity norm to compute the difference between the predicted and true value. To contextualize these differences, maximum heights obtained by the simulations take values between 0.21 m and 3.47 m, and arrival times vary between 2565 s and 5504 s. For both problems, the 16,000 samples computed are randomly split in 12,000 samples for the training set, 2000 for the validation set, and 2000 for the test set. The training or train set is the portion of data used to fit the NN model. The validation set is used to improve the model; it is used to evaluate the model at the same time the model is being trained, but the validation data are not used to train the model. The validation set can be seen as a "weak training set". Finally, the test set works as new data for the already trained model and is used to assess model accuracy. The best model is the one that minimizes the error in infinity norm, in our case for the maximum wave height or the arrival time, depending on the problem.

We used the Keras library [21] to implement and train the NN models proposed in this work.

### 4.1. Models for Maximum Heights

A large set of single neural networks were trained in order to predict the maximum height of a tsunami at six points, and the resulting predictions were tested. These points are located around the coast between Chipiona and Cádiz, see Section 2. Among the long list of models trained, in this section we include 24 models that are summarized in Tables 4–6. The description and results of some selected trained models will be presented in the next section.

**Table 4.** Results of different models for maximum heights with the sigmoid activation function in the output layer.

| $N°$ | Layers | Units | Activation | Batch Size | LR | Loss | Max. Error |
|------|--------|-------|------------|------------|-------|-----------|------------|
| 1 | 6 | 64 | tanh | 256 | 0.002 | Huber(0.1) | 0.092 |
| 2 | 6 | 100 | tanh | 256 | 0.001 | Huber(0.1) | 0.080 |
| 3 | 6 | 100 | tanh | 512 | 0.001 | Huber(0.1) | 0.110 |
| 4 | 6 | 100 | tanh | 256 | 0.002 | Huber(0.1) | 0.088 |
| 5 | 7 | 64 | tanh | 256 | 0.002 | Huber(0.1) | 0.112 |
| 6 | 7 | 100 | tanh | 256 | 0.002 | Huber(0.1) | 0.104 |
| 7 | 7 | 200 | tanh | 256 | 0.002 | Huber(0.1) | 0.122 |

**Table 5.** Results of different models for maximum heights with the sigmoid activation function in the output layer and the *ReduceLROnPlateau* callback. Blue rows are models with good results that will be use in ensemble models.

| $N°$ | Layers | Units | Activation | Batch Size | LR | Loss | Max. Error |
|------|--------|-------|------------|------------|-------|-----------|------------|
| 8 | 5 | 64 | tanh | 256 | 0.002 | Huber(0.1) | 0.107 |
| 9 | 5 | 100 | tanh | 256 | 0.002 | Huber(0.1) | 0.083 |
| 10 | 6 | 64 | tanh | 256 | 0.002 | Huber(0.1) | 0.079 |
| 11 | 6 | 100 | tanh | 128 | 0.002 | Huber(0.1) | 0.096 |
| 12 | 6 | 100 | tanh | 256 | 0.001 | Huber(0.1) | 0.092 |
| 13 | 6 | 100 | tanh | 256 | 0.002 | Huber(0.1) | 0.070 |
| 14 | 6 | 100 | tanh | 256 | 0.002 | mse | 0.087 |
| 15 | 6 | 100 | tanh | 512 | 0.002 | Huber(0.1) | 0.083 |
| 16 | 7 | 100 | tanh | 256 | 0.002 | Huber(0.1) | 0.070 |

**Table 6.** Results of differents models for maximum heights with the *ReduceLROnPlateau* callback but without sigmoid activation function in the output layer. Blue rows are models with good results that will be use in ensemble models and the green row is the individual model selected.

| $N°$ | Layers | Units | Activation | Batch Size | LR | Loss | Max. Error |
|------|--------|-------|------------|------------|-------|-----------|------------|
| 17 | 6 | 64 | tanh | 256 | 0.002 | Huber(0.1) | 0.088 |
| 18 | 6 | 100 | tanh | 256 | 0.002 | Huber(0.1) | 0.081 |
| 19 | 7 | 64 | tanh | 256 | 0.002 | Huber(0.1) | 0.081 |
| 20 | 7 | 100 | tanh | 256 | 0.001 | Huber(0.1) | 0.072 |
| 21 | 7 | 100 | tanh | 256 | 0.002 | Huber(0.1) | 0.068 |
| 22 | 7 | 100 | tanh | 256 | 0.002 | mse | 0.078 |
| 23 | 7 | 200 | tanh | 256 | 0.002 | Huber(0.1) | 0.067 |
| 24 | 7 | 200 | tanh | 256 | 0.001 | Huber(0.1) | 0.072 |

In Figure 6, the maximum height in the high-resolution mesh of 40 m for the reference scenario in Table 3 is depicted.

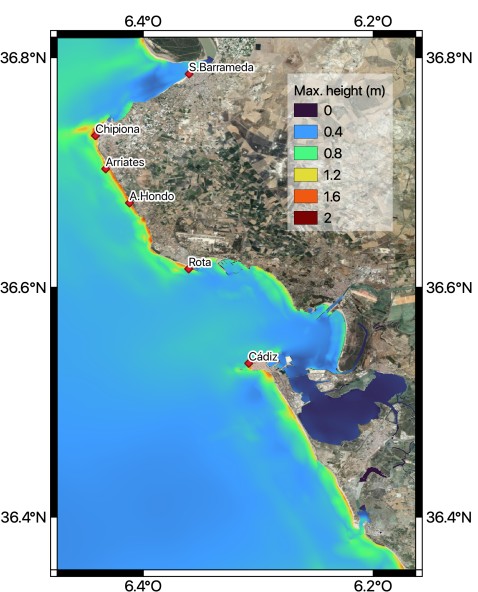

**Figure 6.** Maximum height for the reference event at segment CA01. The spatial resolution is 40 m.

For each model presented, the inputs (the nine Okada parameters) were normalized to [0, 1]. The output layer has six neurons and the output was normalized to [0.02, 0.98]. In most of the models presented, the *Huber* loss function [21] with a threshold, $\delta$, equal to 0.1 is considered. The *Huber* loss function is a piecewise function quadratic for small values of the residuals, and linear for large values, with equal values and slopes where the absolute value of the residuals equals the threshold. In other words, the *Huber* loss function is defined as the mean squared error loss function (*mse*) for values smaller than $\delta$ and as the mean absolute error (*mae*) loss function for values higher than $\delta$. The reason for this partition is to handle outliers. Mean squared error is smooth around 0, not increasing the error of predictions that are in this range, while mean absolute error has different gradients next to 0 and it may start oscillating with small errors. In addition, *mse* is more influenced by extremes than *mae*. The choice of the threshold $\delta$ becomes important depending on what is considered an outlier.

To clarify this explanation, we provide the *Huber* loss formula:

$$L_\delta(y, \hat{y}) = \begin{cases} \dfrac{1}{2}(y - \hat{y})^2, & if \qquad |y - \hat{y}| \leqslant \delta \\[2em] \delta(|y - \hat{y}| - \dfrac{1}{2}\delta), & otherwise \end{cases}$$

The learning rate is a configurable hyperparameter, used in the training of neural networks, that has a small positive value, in the range between 0.0 and 1.0, and it is one of the most crucial parameters [22]. The step size or the *learning rate* is the amount that the weights are updated during training. In the NN models described here, a learning rate of 0.002 returned better results than other learning rates. Models with a learning rate of 0.001 are also included. Regarding the batch size [18] used in the models presented (the batch size is the number of samples processed before the model is updated), different batch sizes were tested, obtaining the best results with a value of 256 samples. Models with batch sizes of 128 and 512 are also presented.

The set of models that are presented were organized into three subsets, briefly described in Tables 4–6. All the models presented use the *Adam* algorithm as optimizer [18] and the *early stopping* callback [21]. The models in Table 4 have a sigmoid as activation function [17] in the output layer. The *ReduceLROnPlateau* callback [21] was added in models in Table 5. It improved the results. Finally, models without the sigmoid activation function in the output layer were developed. Results are presented in Table 6.

Other popular activation functions, such as *relu* [17], different types of initialization weights, and different regularization techniques [15], such as *Dropout*, *L1*, *L2*, and *Batchnormalization*, were also used, but in no case did the results improve.

Some of the models have a maximum height absolute error close to 7 cm and, in particular, model 23 has a maximum error of 6.7 cm or 6.8 cm for model 21. As model 23 presents the smaller value for the maximum, we will retain this model when using a single model NN. Other models, which also return good results (marked in blue), will be used in ensembles. The single models selected for the ensemble modeling do not have to be all of the models presenting the smaller maximum errors; it is important to select different models performing well in different situations. This justifies the choice of model 19.

### 4.2. Models for Arrival Times

In the same way as the NN, for predicting the maximum height at the forecast points, a second set of models was trained for the arrival times at the same six forecast points. In Figure 7, the graphic for the simulated arrival times for the reference simulation is shown.

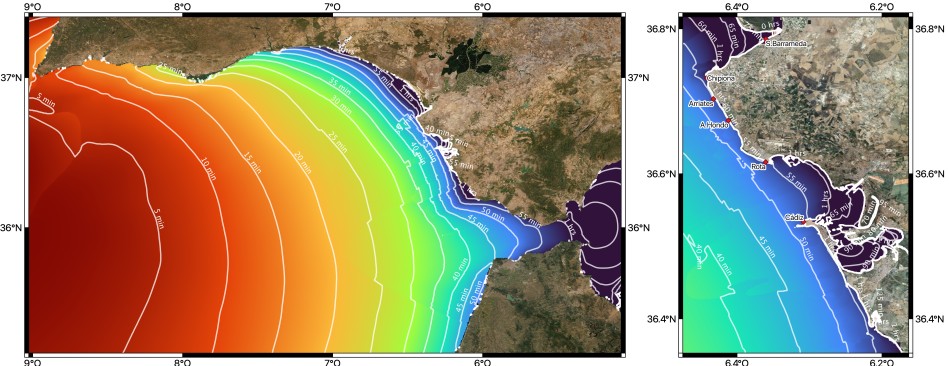

**Figure 7.** Arrival times split by minutes for the reference event. The spatial resolutions are 320 m (**left**) and 40 m (**right**).

Inputs were normalized to [0, 1]. The output layer has six neurons, and in this case, the output was normalized to [0.05, 0.95]. The models in Table 7 have no activation function in the output layer and use the *Adam* algorithm as optimizer. *Early stopping* and *ReduceLROnPlateau* callbacks are used, too. Models without *ReduceLROnPlateau* callback were developed and provided worse results. Table 8 presents these results.

**Table 7.** Results of differents models for arrival times with *ReduceLROnPlateau* callback. Green row is the individual model selected.

| $N°$ | Layers | Units | Activation | Batch Size | LR | Loss | Max. Error |
|------|--------|-------|------------|------------|-------|------------|------------|
| 1 | 5 | 100 | tanh | 256 | 0.001 | Huber(0.5) | 331.82 |
| 2 | 5 | 100 | relu | 256 | 0.001 | Huber(0.5) | 454.34 |
| 3 | 6 | 64 | tanh | 256 | 0.001 | Huber(0.5) | 293.95 |
| 4 | 6 | 100 | tanh | 256 | 0.001 | Huber(0.5) | 238.74 |
| 5 | 6 | 100 | tanh | 256 | 0.002 | Huber(0.5) | 358.62 |
| 6 | 6 | 100 | tanh | 512 | 0.001 | Huber(0.5) | 295.57 |
| 7 | 6 | 100 | tanh | 256 | 0.001 | mse | 272.24 |
| 8 | 6 | 100 | relu | 256 | 0.001 | Huber(0.5) | 319.48 |
| 9 | 7 | 100 | tanh | 256 | 0.001 | mse | 342.91 |
| 10 | 7 | 100 | tanh | 256 | 0.001 | Huber(0.5) | 325.90 |
| 11 | 7 | 200 | tanh | 256 | 0.001 | Huber(0.5) | 307.23 |

**Table 8.** Results of different models for arrival times without *ReduceLROnPlateau* callback.

| $N°$ | Layers | Units | Activation | Batch Size | LR | Loss | Max. Error |
|------|--------|-------|------------|------------|-----|------|------------|
| 12 | 5 | 100 | tanh | 256 | 0.001 | Huber(0.5) | 372.29 |
| 13 | 6 | 64 | tanh | 256 | 0.001 | Huber(0.5) | 332.11 |
| 14 | 6 | 100 | tanh | 256 | 0.001 | Huber(0.5) | 313.74 |
| 15 | 6 | 100 | tanh | 256 | 0.001 | mse | 366.10 |
| 16 | 7 | 100 | tanh | 256 | 0.001 | Huber(0.5) | 346.45 |

Our criterion will be to choose the green model from Table 7 despite that we observed that it has a slight overfitting. This means that the model is very good at predicting the data used to train the model, but is not so good at predicting the data it was not trained on. Therefore, to avoid this, regularization techniques were applied. These techniques will raise the maximum time error to 262 s while greatly reducing overfitting. Regarding the choice of the regularization, we checked, after some tests, that the best option is the following: for model 4, an *L2*-regularization [15] is applied to the model weights in the first layer, and to the output in the last one. More detail can be seen in Section 5.

## 5. Main Results

In this section, we will describe the two neural network models that were selected as single models from the 24 models in the case of forecasting the maximum height and from the 16 models presented in the case of the arrival times. The results obtained with these two models will be analyzed in terms of percentage of predictions within error intervals. In addition, in this section, we will also introduce ensemble methods, as already described. These kinds of methods can help us to improve the performance of the single models. Several ensemble models, using two, three, and four single models, were tested for both problems, and some results are presented.

### 5.1. Single Models for Maximum Heights

In the previous section, all models for the maximum height forecast were shown to produce good results, with maximum errors ranging from 6.7 to 12.2 cm. The best single model among the 24 models developed for the maximum height problem is model 23. For all maximum height models, inputs were normalized to [0, 1] and the output to [0.02, 0.98]. The model producing the smallest maximum error is model 23. This model is composed of seven layers, six hidden layers, and the output layer. Each of the hidden layers has 200 neurons and the hyperbolic tangent function as activation function. As we are predicting at six locations, the output layer has six neurons and it has no activation function. The *Adam* optimizer is used with a initial learning rate of 0.002 (learning rate will be decaying with the *ReduceLROnPlateau* callback). The batch size selected is 256 and the *Huber* loss function is selected with $\delta = 0.1$.

Two callbacks were applied. The *early stopping*, that stops the model when the minimum of the mean of the validation set predicted is not decreasing after 300 epochs, and the *ReduceLROnPlateau*, which reduces the learning rate if the validation loss has not been reduced in 200 epochs, multiplying by 0.8 each time and with a minimum learning rate of 0.0005. It takes 6125 epochs to train the NN.

The left panel in Figure 8 shows the evolution of the error as the model is trained. It can be observed how the callbacks are acting: The choice of the learning rate produces big jumps at the first part of the training and smaller variations occur when the number of epochs increase (larger than 2000). In addition, when the error is no longer decreasing, the *early stopping* concludes the training. In general, to keep training, besides this point, does not necessarily improve the model. In Figure 8, right panel, we can see how many of the predictions are in a given error interval.

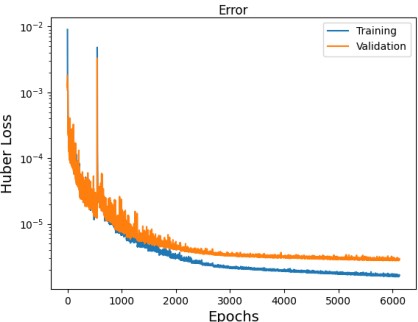
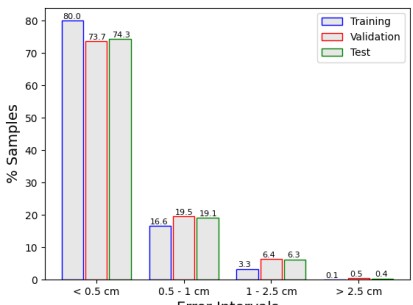

**Figure 8.** Train and validation error during training (**left**) and percentage of predictions in certain error intervals in the maximum height problem (**right**).

Table 9 provides the results of the single neural network selected (model 23). It shows the absolute mean error and the absolute maximum error in each of the points and sets defined before. Very good results are obtained, with an average mean error of 0.38 cm for the test set at the six locations, and a maximum height error of 6.77 cm at Rota.

**Table 9.** Mean and maximum absolute errors for the reference NN model (23) for the predicted maximum heights.

| Location | Mean Error (cm) | | | Maximum Error (cm) | | |
|---|---|---|---|---|---|---|
| | Train | Validation | Test | Train | Validation | Test |
| S. Barrameda | 0.17 | 0.22 | 0.21 | 1.24 | 2.46 | 2.68 |
| Chipiona | 0.36 | 0.44 | 0.42 | 4.99 | 3.50 | 4.75 |
| Arriates | 0.35 | 0.41 | 0.40 | 2.60 | 3.11 | 6.26 |
| A. Hondo | 0.35 | 0.42 | 0.42 | 2.23 | 4.13 | 4.95 |
| Rota | 0.41 | 0.54 | 0.55 | 5.05 | 6.77 | 6.68 |
| Cádiz | 0.25 | 0.31 | 0.30 | 4.33 | 5.30 | 3.46 |
| Average | 0.31 | 0.34 | 0.38 | 3.40 | 4.21 | 4.79 |

### 5.2. Ensemble Models for Maximum Heights

Once the single neural networks have been trained, combinations of several of them can be used to improve the results. We tried several combinations. Two different types of ensemble were attempted. One of them just performs a sampling of the space of possible solutions (weighted average MLP ensembles [20]), and the other uses an optimization process, using any available information to make the next step in the search.

Tables 10–12 provide results of some possible combinations with two, three, and four models. In the column "Weights", we round the weights associated with each model in the ensemble, and the column "Type" refers to the type of ensemble method used, space search or optimized search.

**Table 10.** Ensemble models tested using two single models and the maximum error obtained.

| N° | Models | Weights | Type | Max. Error |
|---|---|---|---|---|
| 1 | 2; 13 | [0.041, 0.959] | space | 0.070 |
| 2 | 2; 13 | [0.116, 0.884] | optimized | 0.071 |
| 3 | 2; 16 | [0.003, 0.997] | space | 0.069 |
| 4 | 2; 16 | [0.069, 0.931] | optimized | 0.070 |
| 5 | 13; 21 | [0.432, 0.568] | space | 0.063 |
| 6 | 13; 21 | [0.637, 0.363] | optimized | 0.062 |
| 7 | 13; 23 | [0.501, 0.499] | space | 0.063 |
| 8 | 13; 23 | [0.655, 0.345] | optimized | 0.063 |
| 9 | 21; 23 | [0.887, 0.113] | space | 0.068 |
| 10 | 21; 23 | [0.528, 0.472] | optimized | 0.067 |

**Table 11.** Ensemble models tested using three single models and the maximum error obtained.

| N° | Models | Weights | Type | Max. Error |
|----|--------|---------|------|-----------|
| 11 | 2; 13; 23 | [0.064, 0.522, 0.414] | space | 0.062 |
| 12 | 2; 13; 23 | [0.142, 0.577, 0.281] | optimized | 0.063 |
| 13 | 2; 14; 23 | [0.034, 0.442, 0.524] | space | 0.063 |
| 14 | 2; 14; 23 | [0.190, 0.550, 0.260] | optimized | 0.071 |
| 15 | 13; 16; 23 | [0.248, 0.248, 0.504] | space | 0.063 |
| 16 | 13; 16; 23 | [0.428, 0.381, 0.191] | optimized | 0.062 |

**Table 12.** Ensemble models tested using four single models and the maximum error obtained. Green row is the ensemble model selected.

| N° | Models | Weights | Type | Max. Error |
|----|--------|---------|------|-----------|
| 17 | 13; 16; 19; 23 | [0.284, 0.358, 0.295, 0.063] | space | 0.060 |
| 18 | 13; 16; 19; 23 | [0.395, 0.352, 0.160, 0.093] | optimized | 0.062 |
| 19 | 13; 16; 21; 23 | [0.261, 0.261, 0.350, 0.128] | space | 0.061 |
| 20 | 13; 16; 21; 23 | [0.368, 0.374, 0.195, 0.063] | optimized | 0.062 |

Several ensemble models combining five single models were tested, but in most cases, one of the weights associated with one particular single model was very close to 0, making this particular model contribution negligible.

As has been shown, several combinations for the ensembles were tried. Model 17 returns the best results, with an mean absolute error of 0.3 cm and a maximum error of 6 cm. This ensemble is composed of four models. Table 13 and Figure 9 allow us to compare the results obtained in this case with the single model selected before (Table 9 and Figure 8). Model performance was substantially improved.

**Table 13.** Mean and maximum absolute errors with the ensemble model 17 for the predicted maximum heights.

| | Mean Error (cm) | | | Maximum Error (cm) | | |
|---|---|---|---|---|---|---|
| Location | Train | Validation | Test | Train | Validation | Test |
| S. Barrameda | 0.13 | 0.17 | 0.16 | 1.12 | 1.71 | 2.31 |
| Chipiona | 0.27 | 0.35 | 0.33 | 5.76 | 3.86 | 4.72 |
| Arriates | 0.24 | 0.29 | 0.29 | 2.44 | 3.33 | 6.00 |
| A. Hondo | 0.25 | 0.31 | 0.32 | 2.50 | 2.97 | 3.27 |
| Rota | 0.29 | 0.40 | 0.40 | 3.27 | 5.71 | 4.40 |
| Cádiz | 0.19 | 0.24 | 0.24 | 4.82 | 5.73 | 4.29 |
| Average | 0.22 | 0.29 | 0.29 | 3.31 | 3.88 | 4.16 |

### 5.3. Single Models for Arrival Times

In Section 4, 17 NN models for the prediction of the arrival time of the tsunami wave were proposed, including the one to which we applied the regularization (see models in Tables 7 and 8). The maximum error at estimating the arrival time ranged from 238.74 s for model 4 to 454.34 s for model 2. Errors in arrival times of about 5 min for an application to an early warning system are acceptable. As in the case of the maximum height prediction problem, the inputs were normalized to [0, 1].

Regarding the single model giving a smaller maximum error (model 4), it has six layers, five hidden layers, and the output layer. Each of the hidden layers has 100 neurons, and the hyperbolic tangent is used as activation function. The output layer has six neurons and no activation function. To reduce the mentioned overfitting from model 4, a soft *L2* regularization on the weights was applied on the first layer. Additionally, a soft *L2* regularization on the output was applied. The *Adam* optimizer was used with a initial learning rate of 0.001 (learning rate will be decaying with *ReduceLROnPlateau* callback).

The batch size selected was 256, and the *Huber* loss function was used with $\delta = 0.5$. The output was normalized to [0.05, 0.95].

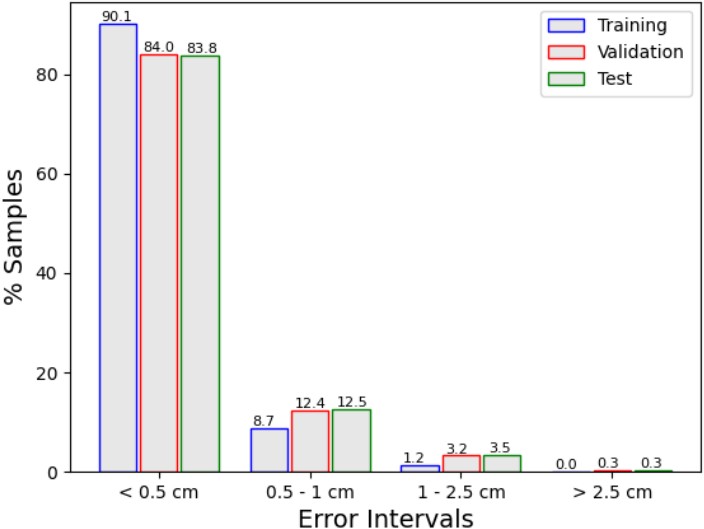

**Figure 9.** Percentage of predictions by error intervals for the height forecasting problem with the ensemble model 17.

Two callbacks were applied. The *early stopping*, that stops the model when the minimum of the mean of the validation set predicted is not decreasing after 500 epochs, and the *ReduceLROnPlateau*, which reduces the learning rate if the validation loss has not been reduced in 180 epochs, multiplying by 0.75 each time and taking a minimum learning rate of 0.0001. It takes 12,975 epochs to train the NN.

Figure 10 (left panel) shows the evolution of the error through the training process and how callbacks are acting. Similarly to what happened in Figure 8, we can observe how the *early stopping* ends the training when the criteria established above has been completed. In the right panel, the percentage of predictions within error intervals is depicted. It can be seen that most of the predictions (95.8% for training, 94.2% for validation, and 94.2% for the testing) have an error below 15 s, which is a very good result.

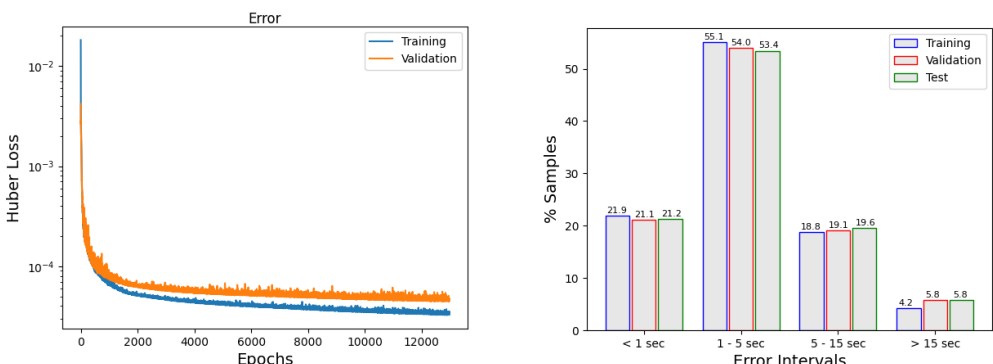

**Figure 10.** Train and validation error during training (**left**) and percentage of predictions in certain error intervals in the arrival times problem (**right**).

Table 14 provides the results for the reference single neural network (model 4 with the above-mentioned regularization). It shows the absolute mean error and the absolute maximum error in each of the six forecasting points and for the three datasets. Very good results are produced, with a mean error of 5 s and a maximum arrival time of 262 s.

**Table 14.** Mean and maximum absolute errors for the reference NN model for the predicted arrival times.

| Location | Mean Error (sec) | | | Maximum Error (sec) | | |
|---|---|---|---|---|---|---|
| | Train | Validation | Test | Train | Validation | Test |
| S. Barrameda | 5.28 | 5.53 | 6.16 | 187.62 | 191.71 | 230.18 |
| Chipiona | 4.40 | 5.16 | 5.26 | 111.60 | 121.96 | 149.80 |
| Arriates | 4.76 | 5.38 | 5.14 | 107.12 | 197.18 | 87.85 |
| A. Hondo | 4.31 | 5.01 | 4.90 | 148.65 | 262.93 | 186.78 |
| Rota | 3.61 | 4.52 | 4.20 | 134.14 | 193.93 | 186.78 |
| Cádiz | 3.89 | 4.74 | 4.35 | 163.15 | 200.51 | 193.04 |
| Average | 4.37 | 5.05 | 5.00 | 142.04 | 194.70 | 172.40 |

### 5.4. Ensemble Models for Arrival Times

Once the 17 single neural networks were trained, combinations of them were tested in order to improve the results. As in the maximum height problem, weighted average MLP ensembles were used. In what follows, we denote as model 17 the one obtained from model 4 after regularization.

Tables 15–17 show the results of some possible combinations using two, three, and four single models.

**Table 15.** Ensemble models tested using two single models and the maximum error obtained. Green row is the ensemble model selected.

| $N°$ | Models | Weights | Type | Max. Error |
|---|---|---|---|---|
| 1 | 3; 4 | [0.041, 0.959] | space | 239.10 |
| 2 | 3; 4 | [0.325, 0.675] | optimized | 246.20 |
| 3 | 4; 7 | [0.901, 0.099] | space | 240.23 |
| 4 | 4; 7 | [0.681, 0.319] | optimized | 248.04 |
| 5 | 4; 14 | [0.981, 0.019] | space | 238.99 |
| 6 | 4; 14 | [0.543, 0.457] | optimized | 258.76 |
| 7 | 4; 17 | [0.362, 0.638] | space | 211.97 |
| 8 | 4; 17 | [0.758, 0.242] | optimized | 228.17 |

**Table 16.** Ensemble models tested using three single models and the maximum error obtained.

| $N°$ | Models | Weights | Type | Max. Error |
|---|---|---|---|---|
| 9 | 4; 7; 17 | [0.350, 0.107, 0.543] | space | 215.42 |
| 10 | 4; 7; 17 | [0.622, 0.228, 0.150] | optimized | 232.47 |
| 11 | 4; 7; 14 | [0.878, 0.061, 0.061] | space | 241.33 |
| 12 | 4; 7; 14 | [0.420, 0.196, 0.384] | optimized | 259.46 |
| 13 | 7; 14; 17 | [0.473, 0.053, 0.474] | space | 265.30 |
| 14 | 7; 14; 17 | [0.275, 0.565, 0.160] | optimized | 291.46 |

**Table 17.** Ensemble models tested using four single models and the maximum error obtained.

| $N°$ | Models | Weights | Type | Max. Error |
|---|---|---|---|---|
| 15 | 3; 4; 14; 17 | [0.227, 0.348, 0.012, 0.413] | space | 227.43 |
| 16 | 3; 4; 14; 17 | [0.190, 0.362, 0.349, 0.099] | optimized | 254.65 |
| 17 | 3; 4; 7; 17 | [0.094, 0.418, 0.050, 0.438] | space | 222.37 |
| 18 | 3; 4; 7; 17 | [0.221, 0.521, 0.134, 0.124] | optimized | 237.99 |

With ensemble techniques, we considered 18 additional models (see Tables 15–17). Among them, we retain ensemble model 7 in Table 15, as it is the one with a smaller mean absolute error (3.90 s) and with a maximum error of 212 s (see Table 18). This ensemble is

composed of the contribution of two single models, models 4 and 17. Table 18 presents the results for the ensemble model 7, which can be compared with the single model results in Table 14. Very good results are obtained, with a mean error smaller than 4 s and a maximum arrival time of 212 s. Figure 11 shows the percentage of predictions within error intervals. It can be seen that most of the predictions (97.8% for training, 96.0% for validation, and 96.2% for the testing) have an error below 15 s, which is a very good result and improves single model results.

**Table 18.** Mean and maximum absolute errors with the ensemble model 7 for the predicted arrival times.

| | Mean Error (sec) | | | Maximum Error (sec) | | |
|---|---|---|---|---|---|---|
| Location | Train | Validation | Test | Train | Validation | Test |
| S. Barrameda | 3.60 | 4.07 | 4.54 | 136.13 | 197.36 | 202.71 |
| Chipiona | 3.25 | 4.12 | 4.26 | 78.73 | 139.93 | 144.83 |
| Arriates | 3.31 | 4.13 | 3.90 | 70.75 | 211.97 | 120.66 |
| A. Hondo | 2.92 | 3.83 | 3.66 | 97.47 | 211.92 | 191.62 |
| Rota | 2.55 | 3.50 | 3.25 | 86.31 | 150.71 | 205.08 |
| Cádiz | 2.61 | 3.89 | 3.46 | 108.51 | 199.13 | 206.63 |
| Average | 3.04 | 3.92 | 3.84 | 81.93 | 185.17 | 178.70 |

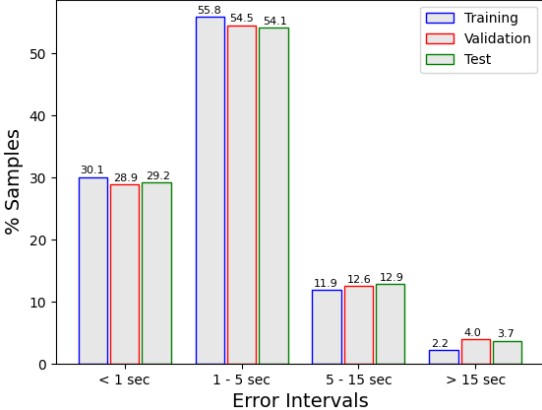

**Figure 11.** Percentage of predictions in certain error intervals in the arrival times problem with the ensemble model 7.

## 6. Towards a TEWS Based on NN Models

The aim of this work is to present a methodological approach that can be used to construct a TEWS based on NN models. The basic building blocks of such a system are described here, in particular NN models for the maximal height forecast and arrival time estimation at a set of forecast points. To have a complete TEW system, for example, on the Atlantic coast of southeastern Spain (Huelva and Cádiz provinces), two extensions must be considered. First, it is necessary to cover the whole coast with forecast points. To do so, our experience leads us to propose a mixed strategy consisting of increasing the number of forecast points in the models we have developed in this paper, on the one hand, and on the other, dividing the coast into segments and training different models on each of these segments. To test the performance of models with more forecast points in a single segment, we will here use 12 points instead of 6 for the CA01 coastal segment. The same set of numerical simulations used to train the six-point models presented in previous sections will be used now to train the 12-point models. An example of this will be described below and the results presented in the Annex. As long as the number of forecast points is not too large, this approach can provide good results; of course, the price to pay is a decrease in performance when the number of forecast points is increased. The second extension required to have a complete TEWS is to train an NN model for every single potential

tsunamigenic source in the area of interest (in our case, in the Northeastern Atlantic area). More specifically, for each active fault, a model with forecast points at each of the segments defined in the coastal stripe (in our case for HU01, HU02, CA01, CA02, and CA03) must be trained.

Finally, we are trying to assess the performance of NN models similar to the two single models that we have retained as reference in previous sections, one for maximum height and another for arrival time, but now trained to output forecasts in 12 instead of in 6 locations. With these 12 points, we could have a coverage of the coastal segment CA01 dense enough for the purpose of a TEWS. Using the same dataset of simulations used to generate all the NN models described in the present study, we trained the two single models with 12 forecasting points for maximum heights and arrival times, and the results obtained are presented in the Annex.

### 7. Conclusions

In recent years, tsunami simulation codes have been developed and improved, returning very accurate results in progressively reduced computing times. Nevertheless, the use in TEWS requires extremely short computing times, not possible yet if high resolution or inundation must be computed. To solve this problem, in this paper, we have proposed the use of deep learning techniques to predict the maximum height and the arrival time to six forecasting points in the coast of Cádiz, using the Horseshoe fault as generating fault.

For each of these two problems, we developed single models and combined some of them to obtain ensemble models that improved the results. The results obtained confirm that deep learning is a useful tool to predict the maximum height and the arrival time of a tsunami at several offshore points near the coast.

As summary:

- The maximum heights obtained by the simulations reach values between 0.21 m and 3.47 m, and the arrival time oscillates between 2565 s and 5504 s.
- The mean absolute error we obtained with a single model for tsunami height is 0.38 cm and the maximum error is 6.77 cm.
- Most of the samples are under 2.5 cm of absolute error for the maximum height.
- Ensemble techniques improve the performance of the results, reducing the maximum error to 6 cm.
- The mean absolute error we obtained with a single model for tsunami arrival times is 5 s and the maximum error is 262 s.
- Most of the samples are under 15 s of absolute error for the arrival time.
- Ensemble techniques improve the performance of the results, reducing the maximum error to 212 s.

The results presented are promising: the estimations at the forecast points provided by the NN models developed are accurate with small errors. The outputs of these models can serve to generate alert levels, which is the final aim of a TEWS, along the coast. To do so, only a few points per coastal segment are required. This justifies the development of six-point models for each coastal segment. Nevertheless, we also generated 12-point models, and the results obtained are presented in the Appendix A. This was performed using the same dataset of numerical simulations. As long as the number of forecast points is not too large, the resulting models also provide good results, although (and obviously) with decreasing performance as the number of points increases.

The next step on the way to have a complete TEWS in this region is to train NN models for assessing maximum heights and arrival times for this same segment, CA01, but for all the potential seismic sources in the area. This means performing on the order of 16,000 numerical simulations for each active fault with the potential of generating tsunamis with impact in the selected coast. Then, and finally, the same process must be repeated for the other four areas defined by the Spanish TEWS in the Atlantic coast of Andalucia, in particular for H01, H02 for Huelva and CA02, CA03 for Cádiz (see Figure 1).

Finally, the results of this study can be extended to other tsunami forecasting problems. For example, the use of convolutional neural networks or recurrent neural networks to simulate the inundated area, or the use of physically informed neural networks (PINNs) to increase the complexity of the physical model (dispersive or non-hydrostatic models).

**Author Contributions:** Conceptualization, J.F.R. and M.d.l.A.; methodology, J.F.R., J.M., M.J.C. and M.d.l.A.; software, J.F.R., M.J.C. and M.d.l.A.; validation, J.F.R.; resources, J.M.; data curation, C.S.-L.; writing initial draft, J.F.R.; writing—review and editing, J.M.; visualization, C.S.-L. and J.F.R.; supervision, J.M. and M.J.C.; project administration, J.M.; funding acquisition, J.M. and M.J.C. All authors have read and agreed to the published version of the manuscript.

**Funding:** This work was funded by "Innovative ecosystem with artificial intelligence for Andalusia 20205" project of CEI Andalucía Tech and University of Málaga, UMA-CEIATECH-05. The numerical results presented in this work were performed with the computational resources provided by the Spanish Network for Supercomputing (RES) grants AECT-2020-1-0009 and AECT-2020-2-0001. Finally, this research has been partially supported by the Spanish Government research project MEGAFLOW (RTI2018-096064-B-C21), ChEESE project (EU Horizon 2020, grant agreement N. 823844), and eFlows4HPC project (funded by the EuroHPC JU under contract 955558 and the Ministerio de Ciencia e Innovación, Spain).

**Institutional Review Board Statement:** Not applicable.

**Informed Consent Statement:** Not applicable.

**Data Availability Statement:** Not applicable.

**Acknowledgments:** We would like to acknowledge the two anonymous reviewers for their very valuable comments that helped us to greatly improve the quality of the manuscript.

**Conflicts of Interest:** The authors declare no conflict of interest.

## Abbreviations

The following abbreviations are used in this manuscript:

| | |
|---|---|
| CAT-INGV | Centro Allerta Tsunami - Istituto Nazionale di Geofisica et Vulcanologia |
| CINECA | Italian Supercomputing Center |
| EDANYA | Ecuaciones Diferenciales, Análisis Numérico y Aplicaciones |
| FTRT | Faster Than Real Time |
| GPU | Graphics Processing Unit |
| HySEA | Hyperbolic Systems and Efficient Algorithms |
| IGN | Instituto Geográfico Nacional |
| NEAM | North Eastern Atlantic and Mediterranean |
| NN | Neural Network |
| ML | Machine Learning |
| MLP | Multi Layer Perceptron |
| TEWS | Tsunami Early Warning System |

## Appendix A. Single Models with 12 Forecasting Points

In order to assess the capability of the NN models developed in this work to produce accurate forecasting when increasing the number of forecast points, we considered extensions of the two single reference models that were proposed in Section 5.1 for the maximum height and in Section 5.3 for arrival times. We trained them with the same set of numerical simulations used to train all the six-point models presented in this work, but now the number of forecast points in the output layer is 12. Table A1 contains the coordinates and depth for the additional forecast points considered, which are interleaved with the six previous points in Table 2. Figure A1 depicts the location of the 12 forecast points.

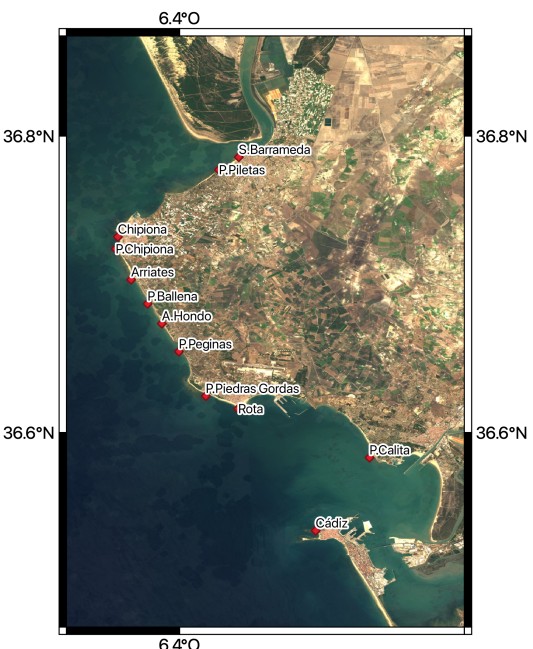

**Figure A1.** Location of the twelve forecast points considered for the 12-point NN models.

**Table A1.** Geographical coordinates of the new locations used as forecasting points. (*) Depth values (in meters) refer to the numerical interpolations.

| Location | Longitude | Latitude | Depth ($*$) |
|---|---|---|---|
| Point 1 (Playa de las Piletas) | $-6.3734°$ | $36.7773°$ | 0.0391 |
| Point 2 (Playa de Chipiona) | $-6.4430°$ | $36.7239°$ | 1.4991 |
| Point 3 (Playa de la Ballena) | $-6.4215°$ | $36.6869°$ | 1.8815 |
| Point 4 (Playa de Peginas) | $-6.4004°$ | $36.6546°$ | 1.5570 |
| Point 5 (Playa de las Piedras Gordas) | $-6.3823°$ | $36.6246°$ | 0.7617 |
| Point 6 (Playa de la Calita) | $-6.2720°$ | $36.58317°$ | 6.6297 |

Figure A2 (left panel) shows the evolution of the error as the model is trained for the maximum height problem. In the right panel, the percentage of predictions within error intervals is depicted. It can be seen that most of the predictions (99.8% for training, 99.2% for validation, and 99.2% for the testing) have an error below 2.5 cm, which is a very good result. This results can be compared with Figure 8 for the six-point model. Table A2 presents the mean and maximum absolute errors for the predicted maximum heights at the 12 forecast points model. This results must be compared with data in Table 9 for the six-point model.

Comparing the data collected in the right panels in Figures 8 and A2, it can be observed that the percentages of predictions with errors below 2.5 cm are quite similar and very high for the three datasets (99.6 vs. 99.2 for the test set, for example), but the errors have increased. For the test set, errors below 1 cm are obtained in 93.4% of the cases for the six-point model vs. 89% for the 12-point model. Comparing Tables 9 and A2, it can be observed that average errors have increased. The maximum average error with the six-point model is 4.79 cm in the test set vs. 5.99 cm for the 12-point model in the validation set. These errors are still acceptable but we must be careful when increasing the number of forecast points.

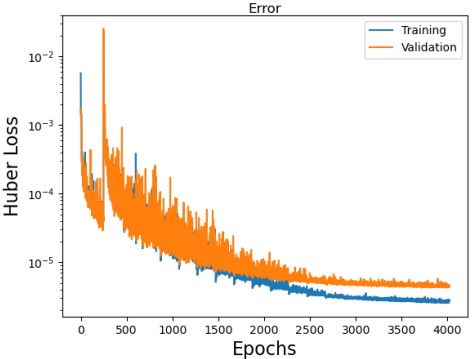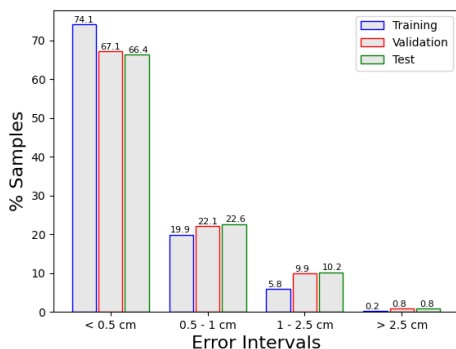

**Figure A2.** Train and validation error during training (**left**) and percentage of predictions in certain error intervals in the maximum height problem (**right**) with 12 forecast points.

**Table A2.** Mean and maximum absolute errors for the predicted maximum heights in 12 forecast points.

| | Mean Error (sec) | | | Maximum Error (sec) | | |
|---|---|---|---|---|---|---|
| Location | Train | Validation | Test | Train | Validation | Test |
| S. Barrameda | 0.20 | 0.24 | 0.25 | 1.95 | 2.57 | 2.32 |
| Point 1 | 0.24 | 0.28 | 0.28 | 2.56 | 2.40 | 2.39 |
| Chipiona | 0.41 | 0.50 | 0.48 | 5.46 | 4.20 | 5.59 |
| Point 2 | 0.29 | 0.37 | 0.36 | 3.43 | 5.78 | 2.78 |
| Arriates | 0.36 | 0.46 | 0.45 | 3.35 | 8.33 | 3.80 |
| Point 3 | 0.43 | 0.54 | 0.56 | 6.69 | 8.79 | 4.31 |
| A. Hondo | 0.40 | 0.50 | 0.52 | 3.29 | 9.24 | 5.80 |
| Point 4 | 0.60 | 0.75 | 0.74 | 5.81 | 5.87 | 8.80 |
| Rota | 0.60 | 0.72 | 0.74 | 6.77 | 6.07 | 7.74 |
| Point 5 | 0.42 | 0.54 | 0.55 | 9.36 | 5.18 | 3.51 |
| Point 6 | 0.26 | 0.32 | 0.33 | 2.30 | 2.83 | 2.65 |
| Cádiz | 0.29 | 0.39 | 0.40 | 5.81 | 10.6 | 7.23 |
| Average | 0.37 | 0.47 | 0.47 | 4.73 | 5.99 | 4.74 |

Moving to the 12-point model for the arrival times, Figure A3 (left panel) shows the evolution of the error as the model is trained, and the right panel presents the percentage of predictions within error intervals. Table A3 provides the results for the arrival times problem for the 12-point model. These results must be compared with data in Table 14 and Figure 10.

Something similar to the maximum height problem occurs. If we compare the percentage of predictions within error intervals in the right panels of Figures 10 and A3, it can be observed that when the error is under 15 s, 1.7% more samples are better predicted with the six-point model, and under 5 s, 16.2% samples obtain a better prediction with the six-point model. All average errors in Table A3 increase if we compare them with the ones in Table 14. Table 14 shows a maximum average error of 194.70 s, while Table A3 shows a maximum average error of 270.37 s.

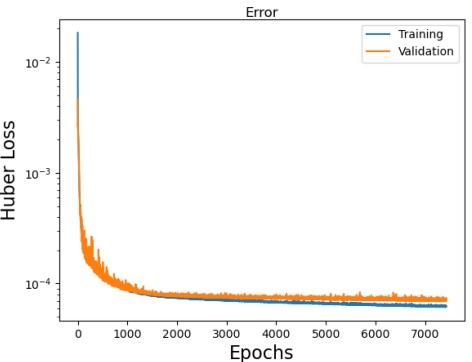 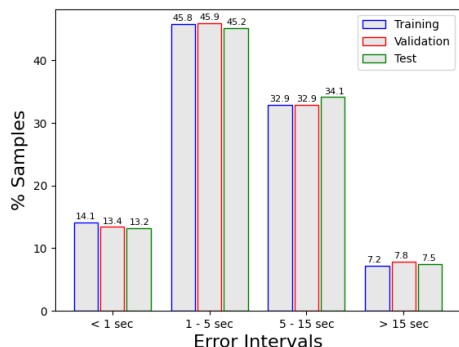

**Figure A3.** Train and validation error during training (**left**) and percentage of predictions in certain error intervals in the arrival times problem (**right**) with 12 forecast points.

**Table A3.** Mean and maximum absolute errors for the predicted arrival times in 12 forecast points.

| Location | Mean Error (sec) | | | Maximum Error (sec) | | |
|---|---|---|---|---|---|---|
| | Train | Validation | Test | Train | Validation | Test |
| S. Barrameda | 7.25 | 7.21 | 7.25 | 212.72 | 219.58 | 195.97 |
| Point 1 | 8.37 | 6.16 | 8.06 | 208.84 | 158.03 | 251.32 |
| Chipiona | 7.30 | 7.48 | 7.61 | 150.81 | 235.02 | 88.53 |
| Point 2 | 7.49 | 7.72 | 7.43 | 151.27 | 141.13 | 140.44 |
| Arriates | 6.50 | 6.89 | 6.91 | 145.69 | 236.01 | 137.27 |
| Point 3 | 6.13 | 6.56 | 6.05 | 131.65 | 127.06 | 119.89 |
| A. Hondo | 7.07 | 7.66 | 7.29 | 209.13 | 270.37 | 125.77 |
| Point 4 | 6.18 | 6.42 | 6.30 | 139.61 | 181.10 | 117.82 |
| Rota | 4.87 | 5.37 | 5.00 | 164.03 | 225.10 | 149.76 |
| Point 5 | 4.57 | 5.21 | 4.72 | 154.55 | 270.18 | 174.92 |
| Point 6 | 4.67 | 5.09 | 5.02 | 202.35 | 194.92 | 262.85 |
| Cádiz | 5.57 | 6.24 | 6.12 | 251.52 | 208.24 | 252.13 |
| Average | 6.33 | 6.67 | 6.48 | 176.84 | 205.56 | 168.05 |

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
