# Peer review of "Use of Neural Networks for Tsunami Maximum Height and Arrival Time Predictions"

_2624-795X, doi:10.3390/geohazards3020017_

Round 1

Reviewer 1 Report

The authors used the concept of a neural network to predict the wave height at six different locations against a pre-defined tsunami condition. The application of numerical models and the use of real-time data are popular methods to inform the temporal and spatial characteristics of tsunami inundation. Although the application of the neural network method is popular in other sciences, it is new to tsunami science. In that sense, the authors' contribution is significant.

Unfortunately, the authors have used only six points to validate their model. In my opinion, the number of points in this study was not sufficient. To show the robustness of their model, they need to consider more points (statistically validated - more than 32 points) where variable bathymetric effects are available (slope and roughness effect).

Line 42 – provide a reference for multi-layer perception.

Line 72- explain the significance of selecting Horseshoe fault for tsunami generation.

Line 80 – to include shallow-water effect in tsunami propagation, the grid size needs to be small (for example 5 m), however, you have selected 40 m, which is quite big. How do you justify this selection?

Fig. 3 shows the temporal variation of sea surface elevation. “Max.Height” is not appropriate.

Table 9 - defines the terms of Train, Validation, Test. You may expand the title of the table.

Fig. 10 What are "Huber Loss" and "Epochs", you need to describe them.

Reviewer 2 Report

The authors show interesting results on prediction of tsunami height and arrival time.

But the results are not relevant for real – time warning.

The range of values of the Okada parameters are approximately corresponding to the intrinsic errors/uncertainties of the various parameters of known fault.

Many seismic zones outside subduction zones have much marger uncertainties on fault parameters. In particular in the North-eastern Atlantic and Mediterranean regions where various types of fault could exist very close the ones from the other. This should be taken into account in such predictions assessment.

The second remark is the parameters of the reference scenario considered (Table 3) are very close to the maximum of the range of values , in particular length (122 – 129) , width (50 – 56), Slip (4.89 – 4.95 = 6 cm), those parameters constrain the travel time and the tsunami height values.

As the reference scenario is close to the maximum scenario, the results of that study are not relevant.

To define the robustness of such method , the reference scenario should be an average scenario (Length 109, Width 46 and Slip 4,0)

What would be the results of such method with this reference scenario ?

To be accepted, the authors should compute and show results with a reference scenario with medium range parameters.

Minor corrections

Table 3 : indicate the scale of parameter (Km, m,..)

L 6 in case of tsunami generated by earthquake.

L21  The two main variables of these systems

L141 The reason ensemble learning is efficient if your machine …

L351 Instituto Geografico Nacional

Figure 3 vertical scale range should be the same

Indicate the 6 forecast points (Figure 3) on Figure 2 and Figure 7

Round 2

Reviewer 1 Report

The authors have addressed my comments.